# A Compact Raster Lensless Microscope Based on a Microdisplay

**DOI:** 10.3390/s21175941

**Published:** 2021-09-03

**Authors:** Anna Vilà, Sergio Moreno, Joan Canals, Angel Diéguez

**Affiliations:** 1Department of Electronic and Biomedical Engineering, University of Barcelona, Martí i Franquès 1, 08028 Barcelona, Spain; sergiomoreno@ub.edu (S.M.); canals@ub.edu (J.C.); angel.dieguez@ub.edu (A.D.); 2Institute for Nanoscience and Nanotechnology-IN2UB, University of Barcelona, Diagonal 645, 08028 Barcelona, Spain

**Keywords:** scan optical microscopy, minute microscope, raster image, microdisplay illumination

## Abstract

Lensless microscopy requires the simplest possible configuration, as it uses only a light source, the sample and an image sensor. The smallest practical microscope is demonstrated here. In contrast to standard lensless microscopy, the object is located near the lighting source. Raster optical microscopy is applied by using a single-pixel detector and a microdisplay. Maximum resolution relies on reduced LED size and the position of the sample respect the microdisplay. Contrarily to other sort of digital lensless holographic microscopes, light backpropagation is not required to reconstruct the images of the sample. In a mm-high microscope, resolutions down to 800 nm have been demonstrated even when measuring with detectors as large as 138 μm × 138 μm, with field of view given by the display size. Dedicated technology would shorten measuring time.

## 1. Introduction

Currently, the benefits of lensless microscopy are not questioned, such as low cost, large field of view (FOV) and no need to focus [1,2]. In short, the technique consists of directly sampling the light transmitted through a specimen located close to a photodetector without the use of any imaging lens. An illumination source (usually a light-emitting diode, LED) lightens the sample through a small pinhole in order to decrease the size of resolvable details [3]. A variety of optoelectronic sensor arrays (such as a charge-coupled device CCD or a complementary metal-oxide-semiconductor CMOS detector) can act as a 2D photodetector. Two main configurations can be categorized in the bright-field operation of lensless microscopy: (i) diffraction-based lens-free microscopy, and (ii) contact-mode shadow imaging. The first configuration relies on computation to revert the effects of the diffraction taking place between the sample and the detector (or an aperture located in between) [4], and in combination with several approaches such as multi-height imaging [5], wavelength scanning [6], sub-pixel shifting [7] or flowing samples [8], sub-micron spatial resolution is possible.

In contrast, the second configuration puts the sample extremely close to the detector (or aperture), so that diffraction is minimized [9]. Light from an illumination source passes through the specimen and casts a shadow on the sensor with unit magnification. With this setup, the resolution is limited by the dimension of the detector pixel and the signal-to-noise ratio becomes independent of the field of view (FOV), allowing unique microscopes that can achieve improved both resolution and FOV simultaneously [10]. Additionally, several strategies have been demonstrated to reduce the effective pixel size at the expense of computational force, such as moving aperture [11] or multi-frame imaging [12].

Concerning high-resolution cameras, Sony, a leader in the world’s imaging sensor market, offers 48-million 0.8 μm-sized pixels onto a 1/2-type (8.0 mm or 0.31″ diagonal) unit [13]. In close competition, Samsung Electronics introduced its latest 0.8 µm mobile image sensor, which comes in 1/1.33″ 108 megapixels (Mp) [14] and produces the industry’s first image sensor with 0.7 μm pixels [15] in a 43.7-megapixel (7968 × 5480) sensor.

CMOS sensors are expected to have smaller pitches and a significantly higher number of pixels due to the ongoing market demand for smaller and higher-performing imaging sensors. However, a qualitative change can be introduced to the traditional lensless configuration by inverting the roles of LED and detector. The application of the principle of reciprocity and methods of Fourier optics for imaging in conventional and scanning microscopes indicates that their behavior is identical [16] and should produce the same image. This means that a scanning optical microscope can be built by measuring in a photodetector the intensity that traverses an object when it is scanned by a point or finite-sized illumination source [17,18]. Conventional and scanning schemes are identical if the light direction is inverted and source and detector interchanged, similarly to what happens with conventional transmission electron microscopy, TEM, and scanning transmission electron microscopy, STEM [19]. Consequently, laser or LED scanning microscopes can be built based on this principle, where the resolution will be limited by the dimension of the light sources and the FOV will correspond to their motion range.

Following this idea, the principles of a microscopy based on a sample scanning that avoids any mechanical movement—of the light source, pinhole or sample—, but just electronically switching every LED in a 2D array in a sequential way, have been recently presented as nano-illumination microscopy, NIM [20,21]. NIM is an effort to achieve low-cost microscopes with resolution beyond the diffraction limit by using nanoscale LED arrays no longer limited by the optical diffraction. NanoLEDs as small as 200 nm were reported [22], opening the possibilities for a practical implementation of this concept.

In NIM setups, the specimen is so close to an LED array that non-diffracted near-field light illuminates it (Figure 1a). The light intensity transmitted through the sample is then captured by a camera as each LED is lit (Figure 1b), to build up a shadow of the sample pixel by pixel, where each LED denotes a pixel (Figure 1c). Although the collected light is diffraction-limited, its origin is the lit LED, whose position is known. As a result, the LED size, and not the diffraction limit, determines the resolution, and the image is reconstructed directly from each shadow without the need of any computational effort. Current nanoLED arrays for NIM are limited to a very low number of LEDs [19,20] which implies a very small FOV of only a few square microns. A similar concept without aiming to obtain a resolution beyond the diffraction limit consists of using a commercial LED microdisplay. This electronically scanned transmission optical microscope, e-STOM, has the advantages of having a large FOV given by the microdisplay area and that it is feasible with current microdisplay technology.

In e-STOM, the observation cavity (where the objects to be observed are placed) can be less than ~1 mm and the active part may only consist of two chips, giving rise to ultra-compact microscopes. Moreover, this setup does not need any optical neither mechanical or computational aid, but only the source array and the photodetector around the sample, making it extremely simple and easy to use, suitable for a broad range of applications were size and weight of the microscope matter. The next generation of e-STOMs will enjoy the benefits of the constantly reduced dimensions of LEDs in microdisplays [23] used in smartphones, tablets, desktop monitors, TVs and augmented/virtual reality devices. In this race, organic LEDs (OLEDs) allow energy-efficient high-contrast microdisplays. Nowadays, major OLED makers provide 0.49″ 1280 × RGB × 720 panels with 8.37 μm pitch (KOPIN [24]), 0.6″ 1280(×3) × 1024 SXGA with 9.3 μm pitch (OLiGHTEK [25]), 0.7″ 1920 × RGB × 1080 Full-HD 7.8 μm pitch 3300 ppi (SONY [26]) or 2048 × 2048 with 9.3 μm pitch (eMagin [27]) microdisplays, for example, and new companies are emerging. However, GaN is a breakthrough in the microdisplay market because of its outstanding performance, brightness and compactness. Major developers of GaN-based microdisplays promise a 0.26″ Full-HD 3.015 μm pixel pitch (Plessey [28]), 0.31″ 720P 5000-dpi 5 μm pitch, 0.22″ Monochromatic 2.5 μm pitch 10k-dpi 1080P or 0.13″ VGA 4 μm pitch 6000 ppi (JBD [29]), for example.

This technique will enjoy the benefits of a single-pixel detector, such as imaging at non-visible wavelengths and with precise timing or depth resolution for a variety of applications [30,31]. Moreover, its theory is known, and even its signal and noise have been modeled as a function of the optical power level, the wavelength of the incident light and the photodiode temperature [32]. This work intends to demonstrate the practical feasibility of an e-STOM. With the proposed configuration, the FOV depends on the lighting microdisplay size, and the resolution is no more given by the detector but by the LED light source. Lighting size smaller than the state-of-the-art microdisplays has been pursued by focusing a light spot onto the specimen by means of a lens or an objective. The lens does not contribute to the image formation, and for this reason the microscope can be considered still lensless. This setup can provide an effective spot dimension given by the central Airy disk of the diffraction pattern as
(1)$$\delta =\frac{1.22\lambda }{\mathrm{NA}},$$
where δ is the spot size, λ the wavelength and NA the numerical aperture of the lens. This e-STOM demonstration may represent the first step of a new concept of microscopy that will result, boosted by the fast-evolving new generations of displays. Additionally, the technique may leverage the displays’ evolution towards microscopy, e.g., for health or the environment.

## 2. Materials and Methods

To achieve these goals, a prototype of an e-STOM has been built with a microdisplay and a camera as the two basic elements. Four light sources (**source 1** to **source 4**, as described next) with progressively reduced dimensions have been tested in this work:Firstly, an UUGL1320 microdisplay (from the Fraunhofer Institute for Organic Electronics, Dresden, Germany [33]), with 720 × 256 OLEDs, 3.6 mm × 1.28 mm area, monochrome green (520–560 nm), diagonal 0.13″, LED size and pitch 5 μm × 5 μm (no spacing between them), providing a brightness of up to 1000 cd/m^2^. A SPI serial interface with a maximum frame rate of 40 fps was used to switch the LEDs on and off one by one.Secondly, a JBD25UMFHDG monochromatic green (520–530 nm) 1920 × 1080 GaN LEDs, 1 μm diameter and 2.5 μm pitch from JBD, Shanghai, China [34] features a 10k dpi resolution with a display area of 4.8 mm × 2.7 mm, providing a luminance of 1.5 Mcd/m^2^. In this case, the images must be uploaded as in a display through an HDMI interface, i.e., the complete image for every frame.For the third light source, a C330TMD-A Mounted Aspheric Lens, ARC: 350–700 nm with f = 3.1 mm and NA = 0.7 (Thorlabs GmbH, Bergkirchen, Germany), was inserted between the first (OLED) microdisplay and the sample, in such a way that the focus plane can be considered as the effective lighting plane. According to Equation (1), this configuration produces an Airy disc of 900–980 nm (owing to the wavelength dispersion). The result of ×10 image reduction is thus a final spot not smaller than 1400–1480 nm with an effective pitch of 500 nm and a FOV for the whole microdisplay of 360 µm × 128 µm.Finally, an Oufemar achromatic objective with ×60 for biological microscopy with tube length of 160 mm and NA = 0.85 was tested between the first (OLED) microdisplay and the sample. This configuration produces an Airy disc of 750–810 nm, originating a final spot of around 830–890 nm with an effective pitch of 83 nm and FOV of 60 μm × 21.3 μm.

The two microdisplays (directly used as **source 1** and **source 2**) have dimensions in the today’s state-of-the-art of their respective (OLED and GaN) technologies. However, in order to assess the suitability of the proposed technique for even smaller illumination sources, this work proposes to add a lens (as in **source 3**) or an objective (as in **source 4**) to focus the lighting onto the sample, such as in a STEM microscope. This configuration can project a demagnified image of the microdisplay onto the sample, acting as an effective smaller light source located directly below the specimen. It is worth to remark that, as the lens is not used for direct imaging, its quality and possible aberrations should not strongly affect the quality of the obtained image, as they would only deform the illumination spot.

Together with every light source, a 728 × 544 pixels monochrome CMOS image sensor (Sony IMX287LLR, from SONY, Tokyo, Japan) with a pixel size of 6.9 μm × 6.9 μm and global shutter was used. The chip was acquired assembled on a printed circuit board from Iberoptics TIS-DMM-37UX287 (from IberOptics, Madrid, Spain) and is controlled via USB3.1.

Three types of samples have been observed with this device. Firstly, a TEM copper grid 3.05 mm in diameter, where 8 μm bars form squares with 54 μm holes (Gilder G400HS copper TEM grid, from Micro to Nano, Haarlem, The Netherlands). Secondly, a custom-made pattern consisting of several features with differently spaced lines, fabricated by lift-off in 40 nm aluminum deposited with an e-beam temescal evaporator (Wordentec, Devon, UK) at a speed of 2 Å/s at room temperature on a fused silica substrate. Thirdly, biological samples such as spirogyra alga or blood cells stained on a cover slide have been observed.

## 3. Results and Discussion

The performances obtained with the different light sources and operation modes are described in this section. As a first result, the final setup is described with some detail, followed by some images and their characteristics.

### 3.1. The Complete Setup

To implement this low-cost portable on-chip imaging platform, the complete system was assembled in an aluminum CNC machined box, 1.26 cm in height, as shown in Figure 2. The small size and light weight of proposed devices such as this one open new field for microscopy, to bring microscopes where nowadays they are not used, e.g., space or field biology, or even to make mobile microscopes able to explore the environment and give rise to distributed microscopy where the sample does not need to be brought into the microscope.

### 3.2. Static and Dynamic Configurations

As an introduction to the main imaging modes in e-STOM, Figure 3 presents some images of a squared TEM grid which has just been dropped onto the OLED microdisplay, acquired by scanning the whole LED array. As the simplest operating mode, a single (static) detector is enough to detect the sample shadows produced by every LED. In contrast, dynamic imaging takes advantage of the 2D detector configuration, collecting the shadow generated by each light source in the detector located in front of it. According to this, Figure 3a was recorded by a detector at the camera center, while in Figure 3b the intensity was measured by a detector that moves to be in front of the lit LED as the scan progresses. An increase in contrast for the whole FOV is visible for the dynamic mode (Figure 3b) in front of the static one (Figure 3a), as the observable area is not limited by the LED beam angle. Despite the fact that static-image equalization may be achieved by increasing intensity for peripheral LEDs, dynamic imaging mode is easier to obtain contrasted images for the full FOV.

Figure 4a–c show details from the same grid positioned at different distances from the display (i.e., changing its height inside the observation chamber), viewed by static e-STOM. For decreasing LED-to-specimen distance (from left to right in Figure 4), off-axis shadows run more inclined (see Figure 4d–f). They are thus sensed by the active detector as coming from fewer sources, which results in increased resolution in the e-STOM image. At the limit, when the specimen is in direct contact with the light sources, shadows evolve almost horizontally (Figure 4g–i). This indicates that the detector size is not critical in e-STOM if the sample is close to the display chip. Thus, a single and large detector (that averages the measurements from several pixels from the camera) can be chosen in static imaging to reduce noise, minimize diffraction effects and to sense with high dynamic range and good timing performances.

By the other hand, in the general static e-STOM the detector is a smaller size than the whole microdisplay. Then, the specimen position in the vertical dimension determines the FOV and affects both sampling area and step by means of the individual LED-detector cones intersecting the sample. Thus, controlling the sample position allows a slight tuning of the effective magnification, and hence of the minimum resolvable spacing. In fact, if there is some spacing between light sources (pitch larger than size, which is not the case of Figure 4), illuminated areas in the sample may not cover it completely, and the small details located between them (i.e., in the sample blind zones) may not be imaged. In this situation, the optimal height for the specimen to be located at, in terms of sampling the whole FOV without shadow merging, would be the plane where the viewing cones contact one another.

A more detailed demonstration of the effects of the distance between sample and light source can be seen in Figure 5, where a dedicated aluminum-on-glass pattern with features in different dimensions and spacings was viewed. Figure 5a shows the image obtained by the conventional lensless approach, using one single LED of **source 2**, the SONY camera and the object located close to the camera. This image can be compared with the one recorded by static e-STOM with the same configuration, except that in this case the object was directly on top of the microdisplay (Figure 5b). In the upper left corner of each picture the respective measuring configuration is sketched, with the sample in red, display at bottom and detector at top. The image improvement visible in Figure 5b, as compared with Figure 5a, recalls the interest of a deeper analysis of the resolution issues, which is undertaken in the next section.

In summary, e-STOM configuration implies that imaging resolution is determined by the microdisplay sample distance. Shadow merging makes the sample located in contact to the microdisplay be optimal. Nevertheless, this ideal situation can be difficult to achieve, owing to fabrication issues including microdisplay chip passivation. In contrast to what happens in conventional lensless microscopy, resolution is limited by the lighting geometry (LEDs size and pitch).

### 3.3. Resolution

Resolution, defined according to the Rayleigh criterion as the smallest distance between two points on a specimen that can still be distinguished as two separate entities, is here explored in some more detail. The reciprocity principle [15] states that resolution in STOM will be limited by the LED’s dimension, and Figure 5 presents an experimental demonstration of it. The worst resolution in Figure 5, obtained by the conventional lensless approach, can be related to the larger intersection between the sample and the light cone between emitter and detector. As in the conventional lensless approach, the sample is located close to the detector; its pixel size (6.9 µm in the case of Figure 5) limits resolution. In contrast, our approach places the object near the microdisplay, and its LED size (1 and 5 µm for Figure 5b,c, respectively) strongly influences the dimension of distinguishable features. In particular, resolution in Figure 5c is around 8–9 μm, and a geometrical approximation suggests that **source-1** passivation is around 700 μm thick. It is worth mentioning that image b, which is best resolved in Figure 5, has been recorded using a detector area of 20 × 20 pixels, which means 138 μm × 138 μm. Despite imaging with a so large sensor, squares 6.4 μm in size can be perfectly distinguished in Figure 5b, thus demonstrating the independence between sensing area and resolution. It seems that, thanks to the very thin passivation of **source 2**, the sample can be very close to the microdisplay, generating huge shadows and providing resolution influenced by the LED’s configuration.

Consequently, the use of smaller LEDs improves STOM resolution, suggesting that our proposal could be useful to pursuit sub-micron resolution. Figure 6 provides an assessment of the actual resolution improvement that can be obtained by diminishing the light source size. Figure 6a corresponds to concentric Al squares made of lines 6.4 μm thick and spaced 6.4 μm, and in Figure 6b the squares are 1.6 μm thick and spaced 1.6 μm, grouped three by three. In both patterns, there are also thick metal lines to separate blocks. In both Figure 6a,b, where **source 3** is used, the 6.4 μm lines and spaces can be distinguished with black/white contrast, indicating that they can be completely resolved. Otherwise, the 1.6 μm ones can only be discriminated with black/grey variations, suggesting that they are near the equipment resolution limit.

To quantify the overall resolution, the intensity profiles of the central strips of the squares in Figure 6a,b are plotted below the respective enlarged Figure 6c,d. The edge spread function (ESF) has been obtained for every large profile step marked in gray, and the FWHM of the contrast derivative (or the 10–90% variation) of a complete step has been calculated to be of around 1.4 μm in both cases. For comparison, the image and the profile in Figure 6e correspond to the same area as Figure 6d, but they have been obtained by using **source 4**. Improved resolution and sharper profiles are evident in Figure 6e, as assessed by an ESF of around 0.8 μm. These two resolution values agree well with the spot sizes provided by **sources 3 and 4**, calculated to be around 1440 and 860 nm, respectively. Figure 6 corroborates then a direct correspondence between e-STOM resolution and source size.

Finally, resolution in conventional lensless microscopy depends on its camera, and it is related to its pixel dimension, as expected. In this work, e-STOM microscopes have been realized with microdisplays based on OLEDs or on direct-addressable GaN LEDs with hybrid interconnection limited by the LED size. Evolution of LED sizes in this kind of microdisplays will be constrained by the LED driving electronics. Other LED technologies, e.g., matrix-addressable GaN microdisplays do not suffer from this limitation because drivers are in the peripheral area of the LED array, and there are only technological issues to the LEDs dimensions. Each of these displays can be used to increase the resolution of e-STOMs and its variations [35], and even to go beyond the diffraction limit. Moreover, optical demagnification, shorter wavelength, increased numerical aperture and larger magnification than the ones used here might allow sub-micron resolution without any kind of processing. Even meta-lenses might help in reaching diffraction-limited focusing and wide FOV [36], still keeping away the need of any image processing.

### 3.4. Considerations about the Optical Downscaling Setup

Interestingly, the dots in plots of Figure 6c–e put in evidence an effect of downscaling that would not be found in direct view by small LEDs. This is that the sub-size pitch provided by **sources 3 and 4** implies a smaller scan step, thus improving profile accuracy. As explained previously, the microdisplay’s pitch and overall size is downscaled by the lens or objective, but the spot dimension is not demagnified at the same scale because it is limited by diffraction. In fact, the actual spot on the specimen should be mathematically described as the convolution of the reduced LED image with the Airy disc of an ideal point-like source. **Source 3** acts as 720 × 256 spots around 1440 nm in size and 500 nm (sub-size) in pitch. Similarly, **source 4** has a final spot of around 860 nm with an effective pitch of 83 nm. In both **sources 3 and 4**, consecutive spots sequentially switched on and off produce oversampled measurements (as sketched in Figure 7) that do not influence resolution but allow better observation of the specimen details. In particular, Figure 6e contains significantly more points than Figure 6d, while its slope is only slightly steeper, as can be attributed to a smaller spot. Consequently, oversampling improves image accuracy and quality, but not its resolution.

If an effective sub-LED source is used, a plethora of possibilities are open, related to effective light-source dimensions and positions. In particular, spot size is determined by demagnification, and image focusing is quite sensitive to the specific height at which the sample is located and referred to the effective light source, as visible in Figure 8.

According to the reciprocity principle [15], the effective detector in a scanning microscope is analogous to the effective source in a conventional one, and its size also affects the imaging process. In the limit case as the detector becomes point-like, imaging becomes coherent and the microscope would exhibit an overall improvement in imaging performance, as in a confocal microscope [17]. In our proposal, partially coherent light sources, such as MicroLEDs [37] and even OLEDs [38], are focused onto the specimen and the detector is of finite size, which would be equivalent to a partially coherent effective detector. Therefore, our complete setup could provide 3D information of the specimen.

Figure 9 shows at which point this effect is achieved, in the image of a portion of spirogyra alga that evidences its helical distribution of the chloroplasts. The main image was recorded with the conventional e-STOM setup by locating the sample close to the light sources, with our 5 μm OLEDs. With the described with-lens setup, the small depth of focus gives information about the 3D structure of the specimen, as visible in the inset of Figure 9, suggesting that a height sweep could allow 3D sample reconstruction. Moreover, imaging resolution is improved as a consequence of the smaller spot size used.

Although these last results show how this with-lens setup can provide 3D information about the specimen, increase resolution, and even improve image quality thanks to oversampling, it suffers from some limitations. Firstly, alignment becomes an issue, as the source image needs to be focused onto the sample to optimize resolution. Secondly, inserting a lens affects the compactness, light weight, design simplicity and cost-effectiveness that are claimed to provide advances such as better integration in complete systems [1]. While an e-STOM without a lens leads these aspects to their limit, the addition of an intermediate lens with high NA can enlarge the observation chamber by 3 mm and increase the cost in EUR 70, which would not severely degrade its applicability into, for instance, lab-on-a-chip platforms. In summary, compactness/simplicity and resolution/image quality are by now competing in e-STOM, and a balance needs to be reached for every application.

## 4. Conclusions

In summary, this paper describes the implementation of an electronically activated scanning transmission optical microscope, e-STOM, based just on a microdisplay and a photodetector. The sample is placed near the light source, and thus, owing to magnification, microscope resolution is conditioned by the emitter dimension and not by the detector. Thanks to the use of partially coherent light, diffraction becomes irrelevant and no image reconstruction is needed, keeping the scheme as the simplest one without any software requirement. This proposal is an actual chip-size microscope, in contrast with other lensless approximations where lighting and detection are made by chips, but the distance between them is on the order of ×10 larger. Moreover, no pinhole or mechanical element is necessary to move the light source nor the sample, as the microdisplay allows this effective motion just by activating electronically the desired LED. These are major advantages compared with the usual lensless setups.

The addition of an intermediate lens can improve resolution down to diffraction-limited spot sizes, and can provide oversampling, as the adequate demagnification allows the object to be mapped with a sub-pixel step. Of course, this oversampling could also be achieved by moving the light source or the sample in small steps, but using a microdisplay and a lens permits the possibility of switching individual LEDs electronically without any mechanical element, as is convenient for low-cost portable large-FOV microscopes. Sub-micron resolution is achieved without any kind of processing.

Several static and dynamic imaging modes can be used with or without lens, including transmission imaging or sub-spot accuracy, which can improve resolution to sub-LED level. The feasibility of an e-STOM by simply using a microdisplay and a photodetector has been demonstrated and implemented in a compact low-cost simple easy-use device. Standard e-STOM imaging takes a substantial amount of time because every LED needs to be activated to get a whole FOV image. However, imaging time could be significantly reduced in very small microscopes if several LEDs would lighten separated specimen areas and they might be used simultaneously with different camera portions. Stitching the obtained sub-FOV images would permit large FOV without any loss in resolution. Additionally, a dedicated technology is key to provide fast cameras that can be synchronized with the microdisplay and generate images more rapidly. Moreover, the use of multiple LED emitters with a single-pixel detector allows significant performance advantages, such as high dynamic range and sensitivity as well as precise timing resolution, which may be used in new applications such as fluorescence-lifetime e-STOM microscopes, at low cost. 

e-STOM microscopes can be built with OLED, hybridly interconnected GaN arrays to CMOS and matrix-addressable microdisplays. Current size limitations on resolution may not be a problem in coming years when building new e-STOM devices and its variations, due to the projected roadmap for microdisplays.

## Figures and Tables

**Figure 1 sensors-21-05941-f001:**
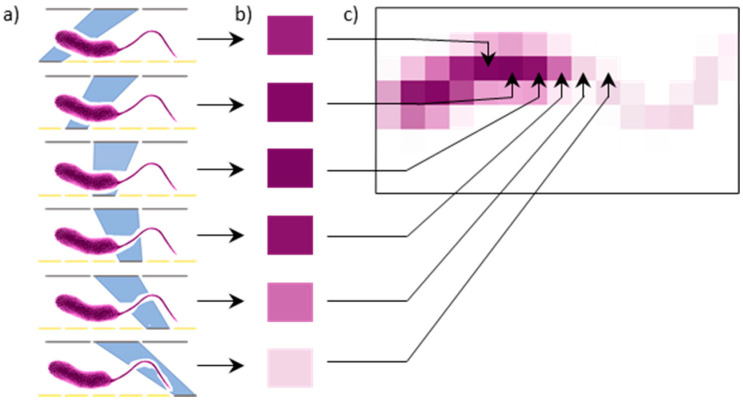
Image formation in NIM and e-STOM. The object is placed on the LED array/microdisplay. Every LED is switched on and off sequentially (**a**), and a detector records the light intensity transmitted through the sample (**b**). The image is formed by tiling every detected intensity as associated with the position of the lit LED (**c**).

**Figure 2 sensors-21-05941-f002:**
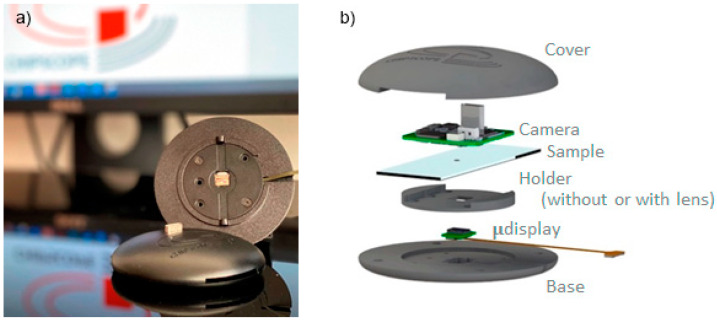
Appearance (**a**) and deployment (**b**) of the proposed e-STOM: inside the oval box there is a detector (**up**), the sample and its holder (**middle**) and a microdisplay (**bottom**). A lens can be located below the sample. Note that the microscope dimension is only limited by microdisplay and detector sizes, and its observation chamber just needs to fit sample and lens, if used.

**Figure 3 sensors-21-05941-f003:**
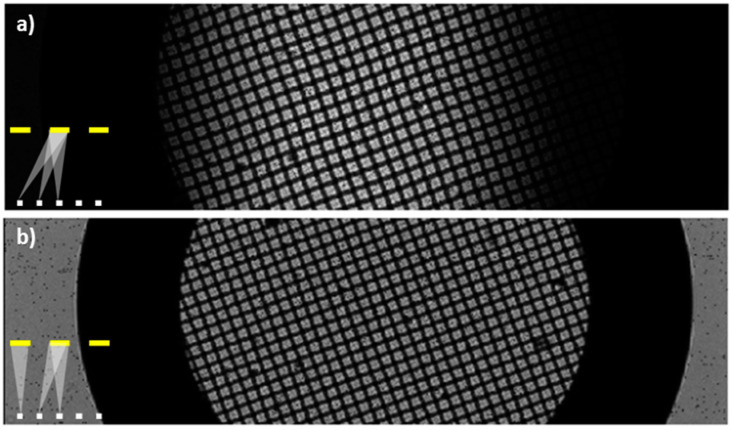
Squared TEM grid placed in contact with **source 1** and viewed with two configurations for the scanned optical microscope: in (**a**) the pixel at the camera center records statically the transmitted light for all the LEDs in the microdisplay, while in (**b**) the light from each activated LED is recorded by the pixel located just in front of it (dynamic measurement). The insets at bottom left show the sketches of the respective operation modes.

**Figure 4 sensors-21-05941-f004:**
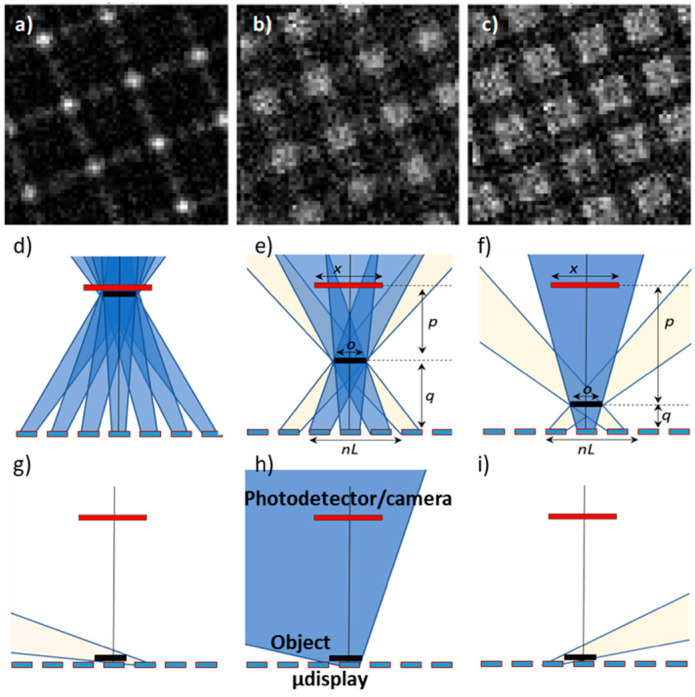
Static images of a squared TEM grid located 4 (**a**), 2 (**b**) and 1 (**c**) mm above **source 1**, in an observation chamber around 10 mm high (i.e., a fixed position of the detector 10 mm away from the display). Schemes of shadows generated by the central LED and its neighbors for different sample heights in (**d**–**f**). The detector senses shadows coming from a larger number of neighbors as the sample is higher, deteriorating the image accuracy. In (**g**–**i**) the case when the sample is in contact with the microdisplay is shown.

**Figure 5 sensors-21-05941-f005:**
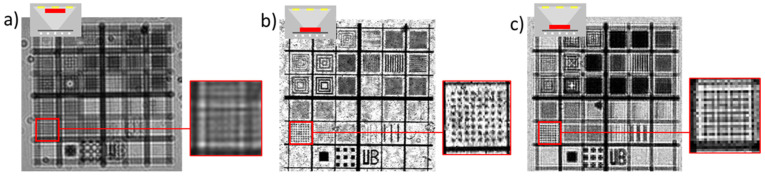
Images of dedicated aluminum-on-glass patterns: lensless with **source 2** (**a**), and static e-STOM images with the sample dropped directly onto **source 2** (**b**) or onto **source 1** (**c**). In all cases the detector is 6.9 μm in size. Insets at top left sketch the respective setups (LEDs at bottom, detectors at top and sample red colored), and enlarged areas show how 6.4-μm squares are resolved.

**Figure 6 sensors-21-05941-f006:**
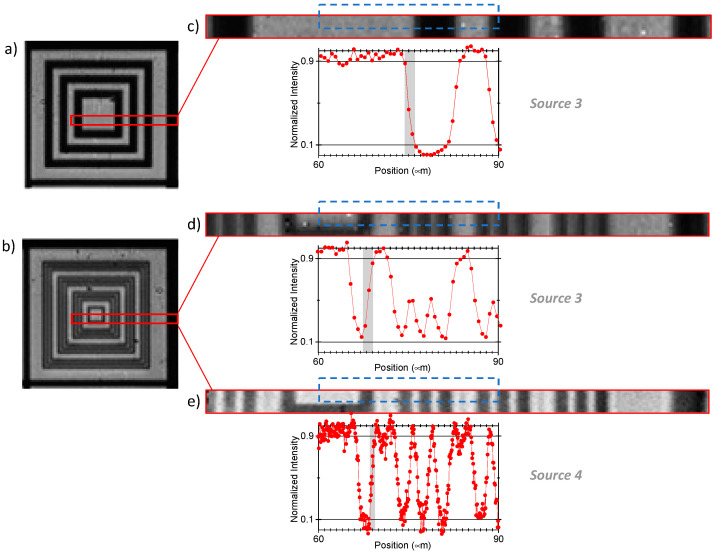
Static e-STOM images of aluminum-on-glass squares. In (**a**) the bars/spacings are 6.4 μm/6.4 μm thick, and in (**b**) they are 1.6 μm/1.6 μm grouped three by three, both imaged with **source 3**. In (**c**,**d**), the respective central strips appear enlarged. In (**e**), the same area of (**d**) is imaged using **source 4**. Below each strip, the normalized intensity profile of the area in the dotted rectangle is plotted, where major (black/white) contrast between the thickest features—used to obtain the ESF—is marked in grey.

**Figure 7 sensors-21-05941-f007:**
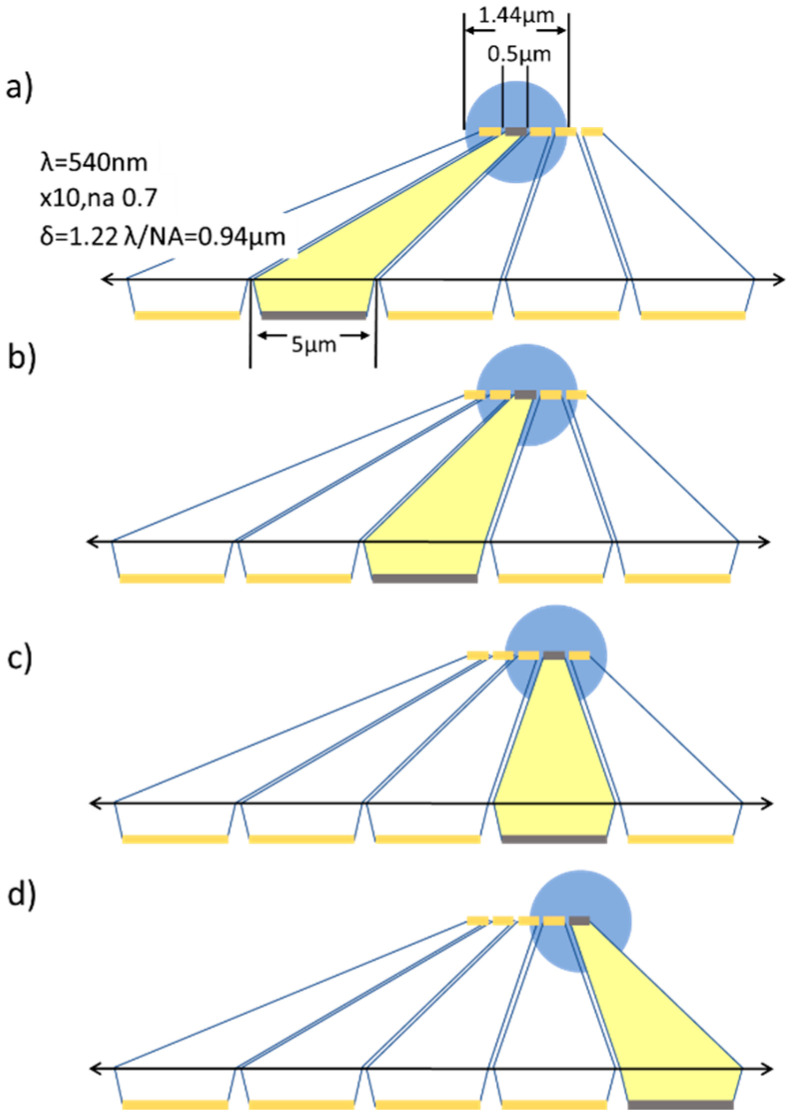
(**a**) Diffracted light spot (at **top**, in blue) produced by downscaling the second LED in the array (in yellow, at **bottom**) by means of a lens in between. The third (**b**), forth (**c**) and fifth (**d**) LEDs downscaled similarly produce spots less spaced than their own dimension, generating oversampled e-STOM images.

**Figure 8 sensors-21-05941-f008:**
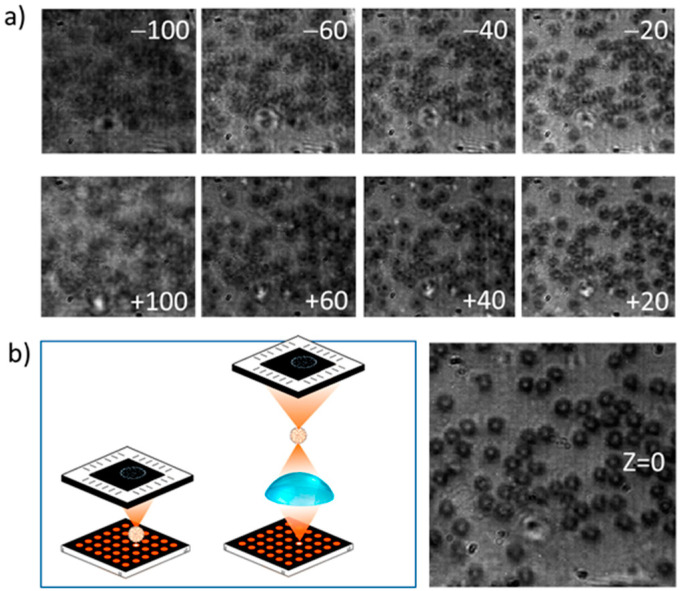
(**a**) Images from smear blood obtained with light **source 3** and the sample located 100, 60, 40 and 20 mm over or below the spot plane. (**b**) Schematics of the without-lens and with-lens setups (at left) and the focused image (Z = 0, at right). All images show the same 100 μm × 100 μm area.

**Figure 9 sensors-21-05941-f009:**
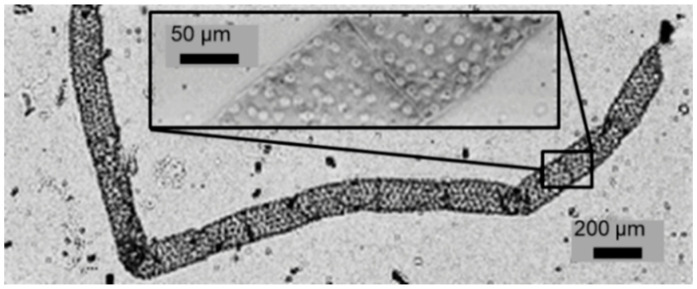
Image of a spirogyra alga recorded with the lens-free e-STOM setup using a 5 μm LED microdisplay (**source 1**). In the inset, image recorded with an intermediate lens reducing ×10 the LED image onto the specimen (**source 3**).

## Data Availability

Not applicable.

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
