# Peer review of "A Compact Raster Lensless Microscope Based on a Microdisplay"

_sensors, 2021, doi:10.3390/s21175941_

Round 1

Reviewer 1 Report

In this manuscript, a smallest practical microscope is demonstrated in which the object is located near the lighting source. In the mm-high microscope, resolutions down to 800 nm have been demonstrated even when measuring with detectors as large as 138 μm x 138 μm, with field of view given by the display size. The writing of the paper is very clear, and the details of the description of the microscope are very detailed, and it can be considered to publish in Sensors.

Author Response

Authors congratulate themselves because their idea seems to have been well understood, and sincerely thank the reviewer for these positive comments.

Reviewer 2 Report

The work concerns a very interesting subject in the field of microscopy. The proposed solution can be used, which can significantly reduce the measurement time. Additionally, such a device can be a portable device. The price of such a device is very low compared to other microscopic devices.
In my opinion, the work has no flaws and should definitely be published. I only ask the authors to correct the descriptions of the drawings to make them more readable.

Author Response

Authors deeply appreciate the reviewer’s positive comments. The description of the figures containing drawings has been corrected to make them more readable:

  • In the case of Fig. 1 from “… The image is reconstructed by associating the detected intensity to the position of the lit LED (c).” to “… The image is formed by tiling every detected intensity as associated to the position of the lit LED (c).”.
  • A “Holder (without or with lens)” label has been added to the sketch in Fig. 2.
  • Caption of Fig. 4 has been changed from “… The detector senses more shadows coming from the neighbors as the sample is higher, deteriorating the image accuracy ...” to “… The detector senses shadows coming from a larger number of neighbors as the sample is higher, deteriorating the image accuracy ...”.
  • As proposed by Reviewer 5, the previous Fig. 5 has been omitted.
  • 5 and 6 (old Figs. 6 and 7) have been rearranged accordingly to the comments of another reviewer. Consequently, their explanation (in the paragraphs immediately above the respective figures) and caption have been revised.
  • The caption of Figs. 7 and 8 (old Figs. 8 and 9) has been slightly modified to improve their readability.

Reviewer 3 Report

Authors present an approach to a lensless microscope. Authors use single-pixel detector and microdisplays and center the resolution of the microscope to the size of the illuminating light source that is placed immediately behind the sample. No major theoretical exploration is made. I strongly advise authors to search for relevant recent bibliography namely from the group of Professor Jesus Lancis of the Universitat Jaume I de Castellón, Spain. The conclusions are sufficiently justified, the limitations are evident, and lines of future investigations are presented. The small size of the microscope system is a good feature to the resolution presented and foreseen but we strongly advise authors to further explore the principle of operation and the new possibilities on the use of sensors single pixel and microdisplay.

Minor corrections to text (missing links, punctuation, spelling)

Author Response

Authors thank the reviewer for her/his comments, which will for sure improve the manuscript quality and clarity. The bibliography has been updated [new references 30–32], and in particular recent publications from professor Jesús Lancis have been found relevant and are cited as [31] and [32].

The basic theory of single-pixel detection is already known (see for instance the new reference 32). However, the detailed principle of operation of this microscope, as well as the new possibilities open on the use of single-pixel detectors together with microdisplays, deserves a deep description in a separate paper. The present manuscript rather aims to demonstrate the practical feasibility of this type of microscope.

Spelling, punctuation, bibliography and typography have been revised.

Reviewer 4 Report

The authors propose a new compact lensless microscope based on a microdisplay. The principle of the method is similar to that of nano-illumination microscopy. Contrarily to other lensless microscopy, the resolution of the proposed method is determined by the light sources. Four light sources are used and the feasibility of the proposed method is verified experimentally in this paper. The manuscript is well organized. I think this paper can be considered for publication in Sensors after answering the following questions:

(1) The diffuser and up-sampling phase retrieval scheme have been employed in the lensless imaging as in Ref. 1, the resolution is no longer limited by the pixel size of the detector. Why do not compare the proposed method with that in Ref. 1? Moreover, compared with the method in Ref. 1, does the e-STOM improve the resolution? In other words, what is the advantage of the e-STOM?

(2) Since a lens or an objective are employed in the paper to produce an effective smaller light source, I think it is better to provide a slightly more detailed version of Fig. 2. In addition, while the intermediate lens is used in the light source, does the system still keep compact?

(3) How does the detector move to be in front of the lit LED during dynamic measurement without the positional misalignment? Does the moving component be required?

(4) A USAF resolution target can be used to test the resolution of the method more directly.

Besides,

(1) In the first Para, Page 1, I guess the ‘such as a charge-coupled CCD device’ here should be ‘such as a charge-coupled device CCD’.

(2) What so the authors mean by saying ‘…making it extremely simple and easy to use for a broad range of applications were size and weight of the microscope matter.’ in the first Para, Page 3?

(3) In Fig. 5, please add scale bars.

Author Response

Authors sincerely appreciate the reviewer’s comments, very useful to improve the manuscript.

Our proposal is an actual chip-size microscope, in contrast with other lensless approximations where lighting and detection are made by chips, but the distance between them is ~ x10 larger. Moreover, no pinhole or mechanical element is required to move the light source nor the sample, as the microdisplay allows this effective motion just activating electronically the desired LED. These are major advantages compared to the usual lensless setups, as the one in ref. 1. Moreover, if compactness, light weight, design simplicity and cost-effectiveness are claimed (in ref. 1) to provide advances such as better integration with lab-on-a-chip platforms, our proposed setup leads these aspects to their limit, as is now commented in the last paragraph of section 3.4.

As in ref. 1, thanks to the use of partially-coherent light, diffraction becomes irrelevant and no reconstruction is needed, keeping the scheme as the simplest one without any software requirement. In ref. 1, pixel super-resolution algorithms and strategies as oil immersion provide half-pitch resolution of 0.3–0.35 mm over a FOV of 20 mm2 and 2.2 mm over 1800 mm2. These values are clearly better than ours (raw), but at this moment we only pretend to demonstrate the practical feasibility of our approach with commercial elements. Moreover, strategies similar to the ones used to improve lensless resolution can also be used in e-STOM. On the other hand, microdisplay technology is being fuelled by the emergent applications of virtual and augmented reality, and it is expected to evolve quickly and provide soon larger microdisplays with smaller LEDs, what will improve directly the FOV and resolution of the proposed configuration to make it competitive against lensless technology.

To clarify this issue, the following text has been added to the Conclusion: “Thanks to the use of partially-coherent light, diffraction becomes irrelevant and no image reconstruction is needed, keeping the scheme as the simplest one without any software requirement. This proposal is an actual chip-size microscope, in contrast with other lensless approximations where lighting and detection are made by chips, but the distance between them is on the order of x10 larger. Moreover, no pinhole or mechanical element is necessary to move the light source nor the sample, as the microdisplay allows this effective motion just by activating electronically the desired LED. These are major advantages compared to the usual lensless setups.”.

The proposed setup can include a lens with high numerical aperture and thus short focus distance. It can then be located at a small distance of both the light source and the sample, keeping the system still compact (in our case, just 3.1 mm higher). Fig. 2 provides the overall appearance of the proposed e-STOM. As the lens can be fixed to the sample holder, the label “Holder (without or with lens)” has been added to the deployment sketch at right, to clarify this aspect.

In addition, the last paragraph of section 3.4 has been modified to discuss is some more detail the demerits of including a lens (such as less compactness), as follows: “Although these last results show how this with-lens setup can provide 3D information about the specimen, increase resolution, and even improve image quality thanks to oversampling, it suffers from some limitations. Firstly, alignment becomes an issue, as the source image needs to be focused onto the sample to optimize resolution. Secondly, inserting a lens affects the compactness, light weight, design simplicity and cost-effectiveness that are claimed to provide advances such as better integration in complete systems [1]. While an e-STOM without lens leads these aspects to their limit, the addition of an intermediate lens with high NA can enlarge the observation chamber in some 3 mm and increase the cost in 70 €, what would not severely degrade its applicability into, for instance, lab-on-a-chip platforms. In summary, compactness/simplicity and resolution/image quality are by now competing in e-STOM, and a balance needs to be reached for every application.”

There is no mechanical or moving element. Taking advantage of the use of a camera instead of a single-pixel detector, dynamic measurement records the intensity detected by the specific pixel which is in front of the lit LED. The sketch in Fig. 3 shows this procedure, and the text immediately above it (1st paragraph in section 3.2) explains it as follows: “As the simplest operating mode, a single (static) detector is enough to detect the sample shadows produced by every LED. In contrast, dynamic imaging takes advantage of the 2D detector configuration, collecting the shadow generated by each light source in the detector located in front of it.”.

Authors thank the reviewer for this suggestion. However, as a first assessment we preferred to use biological or very simple specimens (such as the TEM copper grid), and our own dedicated pattern. A more detailed analysis of the theory and achievable resolution will be the subject of a future publication, and then probably the USAF target will be used to find resolution.

The sentence has been corrected.

The sentence has been changed to “…making it extremely simple and easy to use, suitable for a broad range of applications were size and weight of the microscope matter.”.

As proposed by Reviewer 5, Fig. 5 has been omitted.

Reviewer 5 Report

In this manuscript, a lensless microscope has been developed by CCD camera and LED micro-display. The merit are low cost, compact, and simple configuration. The authors have investigated effects of approaching sample to the LED and the size of the LED. Moreover, a lens has been used for the downscaling. These are very interesting techniques, and thus this manuscript is publishable. However, there are some comments as follows.  

The lines 231-237 including Fig. 5 can be omitted, because they interrupt the discussion about the effect of sample distance from the LED as shown in Fig. 4 and Figs. 6a/c. In the lines 201-230, the sample distance effect for resolution is discussed. It should be followed by the lines 257-270 and a comparison of Fig. 6a and 6c (the latter can be exchanged for Fig. 6b). In the lines 257-270 or 277-286, the authors should focused on the effects of the sample distance or the LED size for resolution (the latter can be understood by a comparison of Fig. 6c and 6b), respectively. 

Fig. 7 can be rearranged to clarify the combination of the panels (c) and (f) for (a), and the panels (d/e) and (g/h) for (b). In the panels (c-e), the position corresponding to the panels (f-h) should be represented by a dotted square and so on.  

In the subsection 3.4, a lens is used in the "lensless" microscope. The authors should discussed not only the merit, but also the demerit in detail. 

In the lines 343 and 347, the reference numbers may be missing. 

Author Response

First of all, authors congratulate themselves because their idea seems to have been well understood, and appreciate the reviewer’s comments, which make the manuscript quality to increase so much.

The text has been arranged according to the reviewer’s suggestions. Lines 231-237 and the previous Fig. 5 have been omitted. The old Fig. 6 (now 5) is rearranged and used to analyse the effects of the sample distance (by comparing Figs. 5.a and b) and of the LED size for resolution (by comparing Figs. 5.b and c) separately in sections 3.2 and 3.3, respectively. The body text has been changed accordingly.

Fig. 6 (old Fig. 7) has been rearranged as suggested by the reviewer, and its caption and explanation have been readapted for more clarity.

Comments about demerits have been added at the end of section 3.4 as follows: “Although these last results show how this with-lens setup can provide 3D information about the specimen, increase resolution, and even improve image quality thanks to oversampling, it suffers from some limitations. Firstly, alignment becomes an issue, as the source image needs to be focused onto the sample to optimize resolution. Secondly, inserting a lens affects the compactness, light weight, design simplicity and cost-effectiveness that are claimed to provide advances such as better integration in complete systems. While an e-STOM without lens leads these aspects to their limit, the addition of an intermediate lens with high NA can enlarge the observation chamber in some 3 mm and increase the cost in 70 €, what would not severely degrade its applicability into, for instance, lab-on-a-chip platforms. In summary, compactness/simplicity and resolution/image quality are by now competing in e-STOM, and a balance needs to be reached for every application.”

Although we have no problems to see these reference numbers, [15] and [17] have been manually inserted at the right positions.

Round 2

Reviewer 3 Report

The authors revised the paper taking into account the reviewers'  comments and the paper may be published in current form.

Reviewer 5 Report

The authors have revised the manuscript properly.